# *Cuscuta* Species Identification Based on the Morphology of Reproductive Organs and Complete Chloroplast Genome Sequences

**DOI:** 10.3390/ijms20112726

**Published:** 2019-06-03

**Authors:** Inkyu Park, Jun-Ho Song, Sungyu Yang, Wook Jin Kim, Goya Choi, Byeong Cheol Moon

**Affiliations:** Herbal Medicine Resources Research Center, Korea Institute of Oriental Medicine, Naju 58245, Korea; pik6885@kiom.re.kr (I.P.); songjh@kiom.re.kr (J.-H.S.); sgyang81@kiom.re.kr (S.Y.); ukgene@kiom.re.kr (W.J.K.); serparas@kiom.re.kr (G.C.)

**Keywords:** *Cuscuta*, parasitic plant, plastid, microscopic analysis, herbal medicine

## Abstract

The genus *Cuscuta* (Convolvulaceae) comprises well-known parasitic plants. *Cuscuta* species are scientifically valuable, as their life style causes extensive crop damage. Furthermore, dried seeds of *C. chinensis* are used as a Korean traditional herbal medicine. Despite the importance of *Cuscuta* species, it is difficult to distinguish these plants by the naked eye. Moreover, plastid sequence information available for *Cuscuta* species is limited. In this study, we distinguished between *C. chinensis* and *C. japonica* using morphological characterisation of reproductive organs and molecular characterisation of chloroplast genomes. The differences in morphological characteristics of reproductive organs such as style, stigma, infrastaminal scale, seed shape and testa ornamentation were useful for distinguishing between *C. japonica* and *C. chinensis*. Analysis of chloroplast genomes revealed drastic differences in chloroplast genome length and gene order between the two species. Although both species showed numerous gene losses and genomic rearrangements, chloroplast genomes showed highly similar structure within subgenera. Phylogenetic analysis of *Cuscuta* chloroplast genomes revealed paraphyletic groups within subgenera *Monogynella* and *Grammica*, which is consistent with the APG IV system of classification. Our results provide useful information for the taxonomic, phylogenetic and evolutionary analysis of *Cuscuta* and accurate identification of herbal medicine.

## 1. Introduction

*Cuscuta* L. (dodder) is a genus comprising 170–200 species of parasitic vines belonging to the family Convolvulaceae [1,2]. Plants in the genus *Cuscuta* are clearly distinguishable from those in other Convolvulaceae genera based on the absence of leaves, presence of haustoria and acotyledonous embryo, arrangement of flowers in clusters or short racemes and presence of five-fimbriate scales within the corolla [2]. *Cuscuta* is nearly cosmopolitan in distribution; however, the majority of the species (~75%) are native to North and South America, and other taxa are found only in Asia and Europe [1,2]. Although some dodders (15–20 species) cause economic or ecological damage to crop production worldwide as agricultural, horticultural or exotic pests [3,4,5], more species are endangered or even threatened, requiring conservation efforts [6]. However, it is very difficult to identify *Cuscuta* species because of the drastically reduced or almost absent vegetative organs (leaves and stems) and very small floral organs. Therefore, most studies have focused on the microscopic evaluation of seed [7], pollen [8], stigma [9] and infrastaminal scales [10] and evolution of these traits of *Cuscuta* species. These studies focused on only the selected character states and their evolution. However, to date, comprehensive species-based analysis of morphological and molecular characteristics has not been performed.

Chloroplasts play important roles in photosynthesis, carbon fixation and starch and fatty acid biosynthesis [11,12]. Recently, many chloroplast genomes have been sequenced using high-throughput sequencing technology [13]. Chloroplast genome sequences are helpful for species identification, population genetics, molecular marker development and analysis of phylogenetic relationship, diversity and evolution as well as for discriminating authentic herbal medicinal plants from adulterants using limited universal DNA barcodes [13,14,15]. In general, angiosperm chloroplast genomes exhibit a conserved quadripartite structure with a large single copy (LSC) region, a small single copy (SSC) region and two copies of inverted repeats (IRs) [16]. The chloroplast genome ranges from 100 to 180 kb in size in higher plants and harbours 110–130 genes, including protein-coding genes and genes encoding ribosomal RNAs (rRNAs) and transfer RNAs (tRNAs) [16]. By contrast, chloroplasts in parasitic plants exhibit reduced genome size, lower gene content and gene loss, as a result of environmental adaptation [17,18,19,20]. In the genus *Cuscuta*, complete chloroplast genomes have been reported for five species, including *C. reflexa*, *C. gronovii*, *C. exaltata*, *C. obtusiflora* and *C. pentagona* [19,20,21]. Chloroplast genomes of these species exhibit gene loss, signatures of selection, genomic rearrangement and differences in genome size, gene number and GC content. Comparison of chloroplast genome sequences of *C. exaltata*, *C. obtusiflora* and *Ipomoea purpurea* shows that all *ndh* genes have been lost in *C. exaltata*. Despite highly variable nucleotide substitutions, *C. obtusiflora* has retained the genes involved in photosynthesis and photorespiration, similar to other parasitic plants [17,20]. Chloroplast genome sequence analysis of *C. reflexa* and *C. gronovii* shows the presence of two inversions in the LSC region and one inversion in the SSC region. Furthermore, *C. reflexa* and *C. gronovii* exhibit different degrees of parasitism, with reduction in RNA editing [19]. Although a few complete chloroplast genomes are available in the genus *Cuscuta*, further investigation is needed for high-resolution phylogenetic analysis as well as *Cuscuta* species identification because of differences in the life style of different parasitic species, numerous existing species and rapid gene evolution.

In Korean traditional medicine, dried seeds of *C. chinensis* are used as an important herbal medicine, namely, Cuscutae Semen, which is designated as a medicine in the Korean Herbal Pharmacopoeia [22], and is regulated by the Ministry of Food and Drug Safety because of its pharmaceutical properties [23,24,25]. Only the seeds of *C. chinensis* are considered as authentic Cuscutae Semen in Korea [22]. However, similar *Cuscuta* species such as *C. japonica*, *C. pentagona* and *C. australis* are also distributed in Korea [26,27]. Since seeds of these species are morphologically highly similar to the naked eye, Cuscutae Semen has been misused and/or mixed with seeds from other *Cuscuta* species in Korean herbal markets. Thus, accurate identification of *Cuscuta* species is important for maintaining a uniform pharmacological effect of the herbal medicine.

In this study, we conducted a microscopic analysis of the reproductive organs of *C. japonica* and *C. chinensis* and sequenced the chloroplast genomes of both species. Results showed differences in the morphological traits of floral organs and seeds as well as in the complete chloroplast genomes of *C. japonica* and *C. chinensis*. Comparison of chloroplast genome sequences of *C. japonica*, *C. chinensis* and related species revealed genomic rearrangements. The genus *Cuscuta* showed highly variable genome structure and gene content among various species. However, subgenera *Grammica* and *Monogynella* exhibited conserved chloroplast genome structure. Our results provide valuable data for the accurate identification of *Cuscuta* species on the basis of morphological and genomic features and in-depth insight into the evolution of species within the Convolvulaceae. Furthermore, this study will help preserve the quality of Cuscutae Semen as a valuable herbal medicine.

## 2. Results and Discussion

### 2.1. Morphological Characteristics of Reproductive Organs

Like other parasitic plants [28], *Cuscuta* species have drastically reduced vegetative organs (stems and leaves) and diverse reproductive organs (flowers and fruits). Thus, morphological characteristics of reproductive organs have been used for the taxonomic and systematic analysis of *Cuscuta* species. Here, we conducted a microscopic evaluation of the morphology of reproductive organs of *C. japonica* and *C. chinensis*.

The morphology of reproductive structures, such as style and stigma, of *C. japonica* (Figure 1a) and *C. chinensis* (Figure 1b) was clearly distinct. Flowers of *C. japonica* exhibited one style with cylindrical and ellipsoid stigma (Figure 1a), whereas *C. chinensis* flowers showed two styles with capitate stigmata (Figure 1b). However, at the cellular level, papillate cells in stigma and ovary, and striate cells in style, were observed in both *C. japonica* (Figure 1a) and *C. chinensis* (Figure 1b). Choisy [29] was the first to propose the infrageneric classification of the genus *Cuscuta*, based on the shape of stigmata. Later, Engelmann [30] used style number and stigma morphology for the classification of three major groups, which have been universally accepted by various authors [31,32,33,34] as subgenera: *Cuscuta*, *Grammica* and *Monogyna* (*Monogynella*). Micromorphological analysis also revealed significant differences between *C. japonica* and *C. chinensis* in the fimbriae of infrastaminal scale (IFS) in flowers. The IFS of *C. japonica* flowers comprised mainly glandular cells (Figure 1c), whereas that of *C. chinensis* flowers comprised secretory cells, called internal laticifers (Figure 1d). Recently, Costea et al. [1] proposed a new classification with four subgenera (*Cuscuta*, *Grammica*, *Monogynella* and *Pachystigma*) and 18 sections, based on the phylogenetic framework and floral morphology including IFS types. In the present study, we confirmed that IFS type, style number and stigma morphology were useful for the identification of *C. japonica* (representative of subgenus *Monogynella*) and *C. chinensis* (representative of subgenus *Grammica*). However, there were no major differences in the epidermal cell type of gynoecium (including stigma, style and ovary) between the two species. This high homoplasy of gynoecium cell patterns may have been affected by convergent evolution, like other floral characteristics [1,9,35,36]. Seed shape varied from pyriform to oblate in *C. japonica* (Figure 1e) but was mainly obovoid in *C. chinensis* (Figure 1f). Moreover, *C. japonica* seeds showed rugulated (puzzle-like) testa ornamentation (Figure 1g), whereas *C. chinensis* seeds showed reticulate testa ornamentation (Figure 1h). These differences in reproductive traits such as the number of styles, shape of stigma and IFS and ornamentation of testa were consistently observed in all samples collected from different sites and individuals each species. Seed characteristics have been used for distinguishing various subgenera and sections of *Cuscuta* [7,37]. In this study, seed shape and testa ornamentation clearly separated *C. japonica* and *C. chinensis*. Moreover, testa ornamentation showed less variability at the species level, suggesting that seed surface may be a stable characteristic, unchanged according to geographical conditions, for differentiating between *C. japonica* and *C. chinensis*. This comprehensive comparative analysis of reproductive organs of two *Cuscuta* species revealed that style number, stigma morphology, IFS type, seed shape and testa ornamentation were the most useful and stable characteristics for species identification (Appendix A).

### 2.2. Chloroplast Genome Features of C. japonica and C. chinensis

To distinguish between *C. japonica* and *C. chinensis* on the basis of chloroplast genome features, we sequenced the chloroplast genomes of both species (Appendix A). Paired-end reads of 2.0 and 2.1 Gb were obtained from *C. japonica* and *C. chinensis*, respectively, with trimmed reads of 1.5 Gb in both species. Complete chloroplast genome sequences of *C. japonica* and *C. chinensis* are 121,037 and 86,927 bp in length, with approximately 69× and 354× coverage, respectively. Both chloroplast genomes showed a quadripartite structure. The length of the LSC region is 79,517 bp and 50,572 bp in *C. japonica* and *C. chinensis*, respectively, and that of the SSC region is 8412 bp and 7121 bp, respectively. Inverted repeat (IR) regions (IRa and IRb) are 16,554 bp and 14,617 bp in *C. japonica* and *C. chinensis*, respectively. The complete chloroplast genomes were validated using PCR-based sequencing using sequence-specific primers (Appendix A). PCR products of the four junctions (LSC/IRa, IRa/SSC, SSC/IRb, IRb/LSC) were analysed by comparison with the complete chloroplast genome sequences (Appendix A). We also validated to read mapping on complete chloroplast genomes (Appendix A). The complete chloroplast genomes were of high quality in both species (Figure 2, Table 1). The GC content of *C. japonica* and *C. chinensis* was 38.3% and 37.6%, respectively (Table 1). In general, the GC content of IRs was higher than that of LSC and SSC regions in both chloroplast genomes. Gene content and gene order were substantially different between the two *Cuscuta* chloroplast genomes; the chloroplast genome of *C. japonica* harboured 96 unique genes, including 65 protein-coding genes, four rRNA genes and 29 tRNA genes (Appendix A), whereas that of *C. chinensis* contained 85 unique genes, including 58 protein-coding genes, four rRNA genes and 24 tRNA genes (Table 1, Appendix A). Both chloroplast genomes showed extensive gene loss within the *nadh* gene family. Additionally, the *C. chinensis* chloroplast genome exhibited the loss of eight tRNA genes and greater gene loss in the RNA polymerase gene family than the *C. japonica* chloroplast genome (Appendix A). The number of intron-containing genes was nine in *C. japonica* and four in *C. chinensis* (Appendix A). Codon usage and anticodon recognition patterns of the chloroplast genomes of both *Cuscuta* species were also analysed. Chloroplast genomes contained 19,180 codons in *C. japonica* and 16,729 codons in *C. chinensis*. Codons for leucine and serine were the most abundant in both chloroplast genomes (Appendix A). Value of the relative synonymous codon usage (RSCU) was greater than one for arginine, leucine and serine codons, as expected (Appendix A). The RSCU values represented synonymous codon bias with a high proportion of A or T in the third position, similar to other chloroplast genomes [38].

Chloroplast genomes of *C. japonica* and *C. chinensis* were compared with those of related *Cuscuta* species. Chloroplast genome features of *C. japonica* and *C. chinensis* were consistent with those of other species in the respective subgenera; the chloroplast genome of *C. japonica* was similar to that of *C. reflexa* and *C. exaltata* in subgenus *Monogynella*, whereas the chloroplast genome of *C. chinensis* was consistent with that of species in subgenus *Grammica* (Table 1). Two subgenera, *Monogynella* and *Grammica*, showed dramatically different genome length, gene content, GC content patterns of chloroplast genomes. Compared with the closest species, *Ipomoea nil*, chloroplast genomes of *C. japonica* and *C. chinensis* showed a loss of 17 and 28 genes, respectively (Appendix A). This gene loss affected the chloroplast genome length [17,19,39]. Genes encoding NAD(P)H-dehydrogenase enzymes were absent in the chloroplast genomes of both species. All *ndh* genes have been lost in both *C. japonica* and *C. chinensis*, similar to other *Cuscuta* species [19,20]. The loss of *ndh* genes has been reported in other parasitic plants as well as in other land plants [40,41,42]. The *ndh* genes were first lost during the evolution of parasitism because of relaxed selection pressure [17,19,20]. Because the loss of *ndh* genes reduced the photosynthetic capacity of plants, *Cuscuta* species evolved to obtain nutrients from host plants [17].

We also surveyed simple sequence repeats (SSRs, also known as microsatellites) and tandem repeats in chloroplast genomes of *Cuscuta* species. SSRs are abundant in genomes, and have been widely used in phylogenetic analysis and population genetics [43,44]. A total of 141 and 94 SSRs were detected in *C. japonica* and *C. chinensis* chloroplast genomes, respectively (Appendix A). The number of SSRs per unit length was higher in single copy regions (LSC and SSC) than in IR regions in chloroplast genomes of both species. SSRs in intergenic spacer (IGS) regions and exons were mononucleotide and dinucleotide in chloroplast genomes of both species (Appendix A). Tandem repeats originate from genomic rearrangements and gene duplication events [45]. We detected 13 and five tandem repeats in chloroplast genomes of *C. japonica* and *C. chinensis*, respectively (Appendix A). Numerous tandem repeats (>100 bp) were detected in the *C. japonica* chloroplast genome. These repeats will serve as useful resources for marker development for species identification.

### 2.3. Dynamic Chloroplast Genome Structure of Cuscuta

To determine the variation in chloroplast genome structure in the genus *Cuscuta*, we compared chloroplast genomes of seven *Cuscuta* species (Figure 3). Rearrangements were detected in four regions among the seven species, and chloroplast genome sizes and gene contents were variable. However, gene content, gene order and genome size were similar at the subgenus level. Two inversions 11.4 and 1.1 kb in length, were detected. The 11.4 kb inversion was present within the LSC region and contained 13 genes (*psbE, psbF, psbL, psbJ, petA, cemA, ycf4, psaI, accD, rbcL, atpB, atpE* and *trnM-CUA*), while the 1.1 kb inversion was present in the SSC region and contained two genes (*ccsA* and *trnL-UAG*); these inversions separated *Cuscuta* species at the subgenus level. Compared with five *Cuscuta* species, chloroplast genomes of *C. japonica* and *C. reflexa* harboured a 2.3 kb inversion (containing *trnT-GGU, trnE-UUC, trnY-GUA, trnD-GUC, psbM, petN and trnC-GCA*) within the LSC region. The chloroplast genome of *C. obtusiflora* harboured a species-specific inversion in the SSC region comprising the *trnL-ycf1* gene. Another inversion, 7.1 kb in length, comprised *trnL-UAG*, *ccsA*, *psaC*, *rps15* and *ycf1* genes in *C. obtusiflora*. Chloroplast genomes in the genus *Cuscuta* exhibited structural vitiations, contained well-conserved syntenic regions which was divided in two subgenera (Appendix A). Single copy regions formed seven well-conserved local collinear blocks among the seven *Cuscuta* species, and IR regions were not detected in syntenic regions (Appendix A). Other *Cuscuta*, *C. reflex* and *C. gronovii*, have structural variation, which is consistent with our results [19]. Large-scale rearrangements were identified by comparative analysis of seven *Cuscuta* chloroplast genomes. We suggest that genomic rearrangements of *psbE-trnM* in the LSC region and *ccsA-trnL* in the SSC region arose before divided to subgenera in *Cuscuta*.

Next, we compared sequences of the junctions between single copy regions and IR regions in the seven *Cuscuta* species. The length of IR regions was lower in the seven species than in other angiosperms. Two patterns of contraction of the IR regions separated the seven species at the subgenus level (Appendix A). In chloroplast genomes of *C. japonica*, *C. reflexa* and *C. exaltata* (subgenus *Monogynella*), the IR/SSC border extended into *ycf2*, resulting in a pseudogene in all three chloroplast genomes, with pseudogenisation of *ycf15* in IRa. By contrast, *C. chinensis*, *C. gronovii*, *C. pentagona* and *C. obtusiflora* (subgenus *Grammica*) harboured *rpl2* in the LSC region. The *ycf1* gene was located in the SSC/IRb border region in all *Cuscuta* species, except *C. exaltata*. In *C. obtusiflora*, inversion was detected in the SSC region. The IR contraction was considerably distinct between *Monogynella* and *Grammica*. Nonetheless, similar patterns were observed between the two subgenera. The IR regions are more highly conserved than single copy regions (LSC and SSC) because of copy correction by gene conversion in IR regions in other angiosperms chloroplast genomes [46]. The contraction and expansion of IR regions has led to variation in chloroplast genome length, which is a crucial criterion for evolution [47,48]. This suggests that IRs in *Cuscuta* plastid genomes underwent more contraction than those in chloroplast genomes of other angiosperms, indicating an independent evolutionary process in response to the environment [49]. To identify genetic variation, we analysed the non-synonymous to synonymous substitution ratio (Ka/Ks) using chloroplast genes conserved among *C. japonica*, *C. chinensis* and *Ipomoea nil*. Results showed no positive selection (Appendix A). Most genes showed a high Ka/Ks ratio in *C. chinensis*. However, in *C. japonica*, values of Ks were lower than those in *C. chinensis*. It is possible that chloroplast genes in *C. chinensis* exhibit more diversity than those in *C. japonica*; however, genetic variability in *C. chinensis* is more stable than that in *C. japonica*. This suggests that chloroplast genes in *C. chinensis* are no longer under selection pressure from the environment and/or have obtained a stable host plant.

### 2.4. Phylogenetic Relationship among Cuscuta Species

Chloroplast genomes have been successfully used in numerous phylogenetic studies of angiosperms because of several advantages such as high accuracy and resolution [13,50]. In this study, we determined the phylogenetic relationship of *C. japonica* and *C. chinensis* with other species within Convolvulaceae using chloroplast genome sequences. A total of 50 conserved protein-coding sequences in 18 chloroplast genomes were aligned over a length of 40,764 bp. A phylogenetic tree was constructed using the maximum likelihood (ML) method. Most nodes showed bootstrap (BS) values of 100% and Bayesian Inference (BI) posterior probabilities (PP) of 1.0, except the node of *Ipomoea* species (Figure 4). Two *Cuscuta* subgenera, *Grammica* and *Monogynella*, formed monophyletic groups within *Ipomoea*. *C. exaltata* formed a paraphyletic group with two *Cuscuta* species, *C. japonica* and *C. reflexa*. *C. chinensis* showed a distinct phylogenetic relationship with the other three *Cuscuta* species within subgenus *Grammica*. The reconstructed phylogenetic tree is consistent with previous studies, according to the APG IV system of plant classification [51,52]. We first represent a reconstructed phylogenetic tree in two *Cuscuta* subgenera based on chloroplast protein-coding sequences. The phylogenetic relationship between the two *Cuscuta* subgenera, *Grammica* and *Monogynella*, was strongly supported by high BS and PP values. Moreover, the *Cuscuta* clade comprising subgenera *Grammica* and *Monogynella* is consistent with morphological and micromorphological traits, such as style number, IFS [1] and testa ornamentation [7]. Further studies involving sampling of all four subgenera of *Cuscuta* [1] and trait analysis are required to enhance our understanding of the evolutionary trends and phylogenetic implication of the reproductive traits within Convolvulaceae.

## 3. Materials and Methods

### 3.1. Plant Material

Plant material used in this study was collected from natural populations over 3 years (2014–2016). To confirm the consistency of morphological and micromorphological characteristics, at least two sites were compared for each species (*C. japonica*: six sites, six individuals; *C. chinensis*: two sites, four individuals). Representative fresh vines of *C. japonica* (36°58′34.5″ N and 128°56′47.5″ E) and *C. chinensis* (33°24′29.9″ N and 126°15′09.8″ E) were used for chloroplast genome sequencing. These samples were assigned identification numbers, and voucher specimens were deposited in the Korean Herbarium of Standard Herbal Resources (Index Herbarium code KIOM) at the Korea Institute of Oriental Medicine (KIOM). Plant samples used for morphological, micromorphological and chloroplast genome sequence analysis are listed in Appendix A.

### 3.2. Microscopic Analysis

For reproductive morphological observations, flowers and seeds were examined using a stereomicroscope (Olympus SZX16; Tokyo, Japan) in order to check the fully mature organs. For scanning electron microscopic observations, the dried materials from voucher specimens were rehydrated overnight using the wetting agent Agepon^®^ (Agfa Gevaert; Leverkusen, Germany) and distilled water (1:200) at 37 °C The rehydrated materials were dehydrated in a graded ethanol series, and dried in an automated critical point drier (Leica EM CPD 300; Vienna, Austria). The dried materials were fixed on aluminum stubs with double-adhesive carbon tape. The stubs were then coated with gold (Au) using an ion-sputtering device (Cressington Scientific Instruments 208HR; Watford, UK). The prepared floral organs and seeds were observed using a low voltage field emission scanning electron microscope (JEOL JSM-7000F; Tokyo, Japan) at an accelerating voltage of 15 kV with a working distance of 9–15 mm [53].

### 3.3. Genome Sequencing and Assembly

DNA was extracted using DNeasy Plant Maxi kit (Qiagen, Valencia, CA, USA), according to the manufacturer’s instructions. Illumina short-insert paired-end sequencing libraries were constructed; sequencing was performed using the MiSeq platform (Illumina, San Diego, CA, USA). Chloroplast genome sequences were assembled de novo from low-coverage whole genome sequences. Trimmed paired-end reads (Phred score ≥20) were assembled using CLC genome assembler (ver. 4.06 beta; CLC Inc., Rarhus, Denmark) with default parameters. Principal contigs representing chloroplast genome sequences were retrieved using Nucmer [54], and aligned with the reference chloroplast genome sequences of *C. gronovii* (NC009765) and *C. reflexa* (NC009766). Gaps in sequences were filled using SOAP *de novo* gap closer, based on aligned paired-end reads [55]. To validate correct genome assembly, sequences of four chloroplast junctions (LSC/IR, IR/SSC, SSC/IR, IR/LSC) were confirmed by PCR-based sequencing using sequence-specific primers (Appendix A). Furthermore, chloroplast reads were mapped onto complete genome sequences using BWA ver. 0.7.25 [56].

### 3.4. Genome Annotation and Comparative Analysis

Annotation of genes in the chloroplast genomes of *C. japonica* and *C. chinensis* was performed using DOGMA annotation [57]. Codons and gene boundaries were manually corrected using BLAST searches. The tRNAs were confirmed with tRNAscan-SE 1.21 [58]. Circular maps of the two *Cuscuta* chloroplast genomes were obtained using OGDRAW [59]. GC content and codon usage were analysed using MEGA6 [60]. Local collinear blocks in the chloroplast genome sequences of all seven *Cuscuta* species investigated in this study (*C. japonica*, *C. exaltata*, *C. reflexa*, *C. obtusiflora*, *C. chinensis*, *C. pentagona* and *C. gronovii*) were identified using MAUVE V2.3.1 program [61]. Substitution rates Ka and Ks were estimated with PAML using the yn00 program [62].

### 3.5. Repeat Analysis

SSRs in chloroplast genomes of *C. japonica* and *C. chinensis* were detected using MISA [63], with minimum number of repeats set to 10, 5, 4, 3, 3 and 3 for mono-, di-, tri-, tetra-, penta- and hexa-nucleotides, respectively. Tandem and palindromic repeats (≥20 bp) were identified with minimum alignment score = 50, maximum period size = 500 and identity of repeats ≥90% [64,65].

### 3.6. Phylogenetic Analysis

A total of 19 chloroplast genomes were used in phylogenetic analysis. Of these, 17 chloroplast genome sequences were downloaded from NCBI (Appendix A). *Capsicum annuum* var. *glabriusculum* and *Nicotiana tabacum* were used as outgroups. Molecular phylogenetic trees were constructed using aligned sequences of 50 protein-coding genes with MAFFT [66], and sequences were manually adjusted using Bioedit [67]. The best-fitting model of nucleotide substitutions was determined using the Akaike Information Criterion (AIC) in JModeltest V2.1.10 [68]. The GTR+I+G model was used in both (Appendix A). The ML analysis was performed using MEGA6 [60] with 1000 bootstrap replicates. BI analysis was performed in MrBayes 3.2.2 [69] using the Markov Chain Monte Carlo (MCMC) method, with two independent runs (four chains each) for one million generations. Phylogenetic trees were sampled every 1000 generations, with the first 25% discarded as burn-in. Phylogenetic trees were determined from 50% majority-rule consensus trees to estimate PP values.

## 4. Conclusions

Morphological characteristics and chloroplast genome sequences of *C. japonica* and *C. chinensis* were analysed in this study. The morphology of reproductive organs such as style, stigma, IFS, seed and testa showed considerable differences between *C. japonica* and *C. chinensis*. Chloroplast genomes were highly variable with respect to gene content, gene orientation and GC content, and local variations in genome sequence and structure were observed at the subgenus level. Tandem repeats and SSRs were identified with the aim of developing molecular markers for species identification and authentication of herbal medicines. Inversions were detected in LSC and SSC regions at *Cuscuta* subgenera. Overall, our results will help in the authentication of herbal medicines containing *C. japonica* and *C. chinensis*. Analysis of key morphological/micromorphological characteristics and chloroplast genome sequences of *C. japonica* and *C. chinensis* provides valuable information for species identification, taxonomic classification and evolutionary analysis of *Cuscuta*.

## Figures and Tables

**Figure 1 ijms-20-02726-f001:**
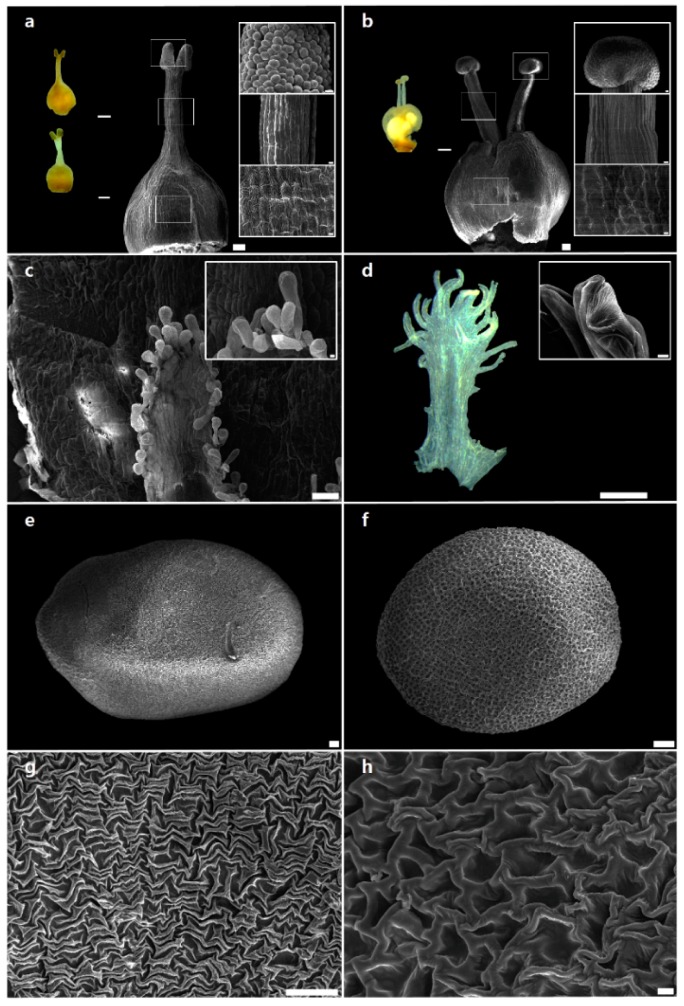
Scanning electron microscope (SEM) and light microscope (LM) micrographs showing the morphology of the major reproductive organs of *Cuscuta japonica* and *C. chinensis*. (**a**) Gynoecium with one style. Small LM micrographs show cylindrical and ellipsoid stigmata (left). SEM micrographs at a higher resolution (right) show cylindrical stigma with papillate cells (top), style with striate cells (middle) and ovary with papillate cells (bottom). (**b**) Gynoecium with two (equal and unequal) styles. Small LM micrographs show unequal styles (left). SEM micrographs at a higher resolution (right) show capitate stigma with papillate cells (top), style with striate cells (middle) and ovary with papillate cells (bottom). Scale bars: 100 μm (**a**,**b**); 1 mm (small LM micrographs of a,b); 10 μm (small SEM micrographs of a,b). (**c**) Fimbriae of infrastaminal scale (IFS). Inset shows IFS with glandular trichomes. (**d**) IFS. Inset shows IFS with secretory cells (internal laticifers). Scale bars: 100 μm (**c**,**d**); 10 μm (small SEM micrographs of c,d). (**e**) Pyriform to oblate seed shape. (**f**) Globose to obovoid seed shape. (**g**) Rugulate (puzzle-like) testa ornamentation. (**h**) Reticulate (net-like) testa ornamentation. Scale bars: 100 μm (**e**–**g**); 10 μm (**h**). (**a**,**c**,**e**,**g**) *C. japonica*. (**b**,**d**,**f**,**h**) *C. chinensis*.

**Figure 2 ijms-20-02726-f002:**
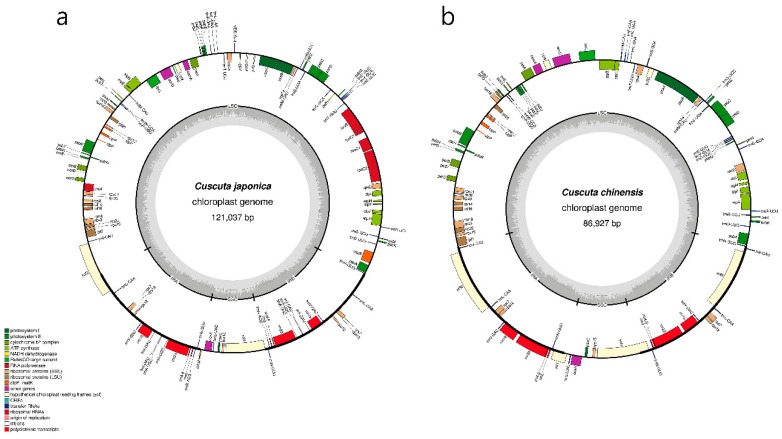
Chloroplast genome maps of *C. japonica* and *C. chinensis*. (**a**) *C. japonica*. (**b**) *C. chinensis*. Genes drawn inside the circle are transcribed clockwise, and those outside the circle are transcribed counterclockwise. Dark grey shading in the inner circle represents the GC content.

**Figure 3 ijms-20-02726-f003:**
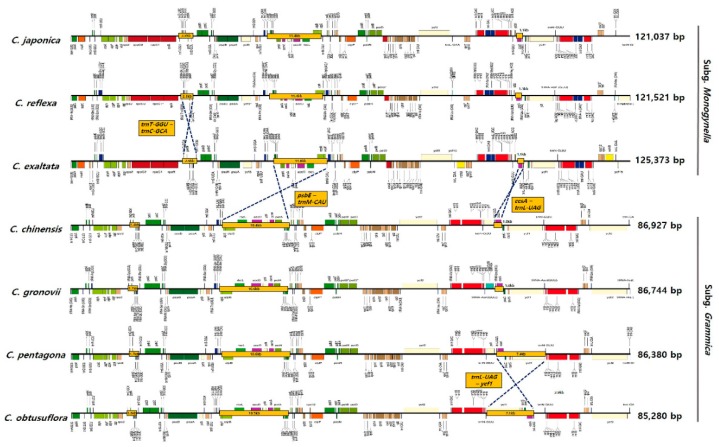
Linear chloroplast genome maps of seven *Cuscuta* species. Linear maps were drawn using OGDRAW. Yellow boxes indicate inversions. Chloroplast genomes of *C. reflexa*, *C. exaltata*, *C. gronovii*, *C. pentagona* and *C. obtusiflora* were downloaded from GenBank.

**Figure 4 ijms-20-02726-f004:**
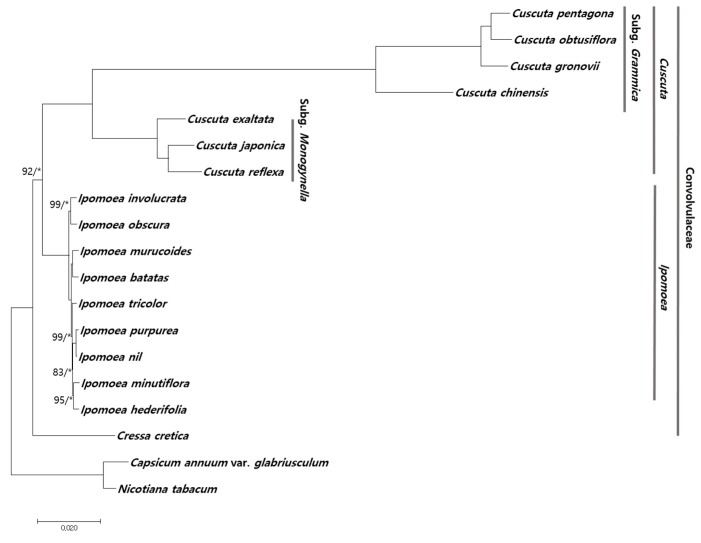
Phylogenetic analysis of *Cuscuta* and *Ipomoea* species. The phylogenetic tree was constructed with the maximum likelihood (ML) method and Bayesian Inference (BI) using 50 conserved protein-coding sequences from seven *Cuscuta* species, nine *Ipomoea* species and *Cressa cretica* within the family Convolvulaceae. *Capsicum annuum* var. *glabriusculum* and *Nicotiana tabacum* were used as outgroups. ML topology is shown with bootstrap (BS) values and Bayesian Inference (BI) posterior probability (PP) values at each node. BS = 100% and PP = 1.0 are not marked. * was shown 1.0 PP values.

**Table 1 ijms-20-02726-t001:** Characterisation of the chloroplast genomes of seven *Cuscuta* species.

	*C. japonica*	*C. reflexa*	*C. exaltata*	*C. chinensis*	*C. gronovii*	*C. pentagona*	*C. obtusiflora*
GenBank accession number	This study	NC009766	NC009963	This study	NC009765	NC039759	NC009949
Chloroplast genome size (bp)	121,037	121,521	125,373	86,927	86,744	86,380	85,280
Large single copy (LSC) region (bp)	79,517	79,468	82,721	50,572	50,973	50,958	50,201
Inverted repeat (IR) region (bp)	16,554	16,741	16,701	14,617	14,354	14,200	14,131
Small single copy (SSC) region (bp)	8412	8571	9250	7121	7063	7022	6817
Total number of genes	96	98	98	85	86	85	86
Number of protein-coding genes	65	65	65	58	58	57	58
Number of rRNA genes	4	4	4	4	4	4	4
Number of tRNA genes	27	29	29	23	24	24	24
GC content (%)	38.3	38.2	38.1	37.6	37.7	37.9	37.8
LSC (%)	36	36.1	35.8	35.9	35.8	36	35.9
IR (%)	45.7	45.7	45.6	42.8	43.1	43.2	43.2
SSC (%)	30.5	30.8	31	28.9	29.5	29.6	29.8

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
