# Peer review of "Cuscuta* Species Identification Based on the Morphology of Reproductive Organs and Complete Chloroplast Genome Sequences"

_ijms, 2019, doi:10.3390/ijms20112726_

Reviewer 1 Report

Park et al., sequenced the chloroplast genomes of two Cuscuta species and did nice comparative genomics as well as morphological analysis. The study was nice, and the manuscript was well written.

Major points:

 L337-340: confusing. First, there was only one phylogeny. Why was GTR+I+G used for two species? Second, it seemed that for such analyses most people may use partitioned phylogenetic reconstruction where different substitution models were given to different partitions of the entire alignment. The authors mentioned that JModelTest was used. Did it mean that the best models of the DNA alignment of each of the 50 genes were determined by JModelTest? If so please show more details in supplemental data. Third, was the third nucleotide in a codon included in analysis? Please correct me if I misunderstood anything.

L170-173: Why was the chloroplast genome of Cuscuta different from other angiosperms? Was that what was shown in L221? Could consider removing L170-173, as it was not until L221 that this point was discussed. Also, more reasons may be needed to support this argument. Likewise, in L234, where was the evidence? Importantly, the case for other plants should be detailed.

The paper 10.1093/jxb/ers391 needs to be cited somewhere. The argument raised in L245-247 may not be the first so references are needed.

Minor points:

1. Figure 2's resolution needs to be improved.

2. In Figure 4. BI of 1.0 was shown as "-". This was a bit confusing to me because i) sometimes this symbol indicates not highly supported nodes ii) that "+" and "-" were shown together makes people wonder whether "-" represents negative results. I understand that the authors plausibly wanted to use it to distinguish between bootstrap values and BI. However, I encourage the authors to revise it or alternatively please tell me if there are other good reasons for it or studies presenting it in the same manner.

3. Figure S7: Genes whose ka == 0 can be shown as they do not affect the calculation of ka/ks (very strong negative selection). But the authors could keep it as it was for this time.

4. The accession number in Genbank for C. pentagona was different between Table 1 and Table S10.

5. L59: exhibit selection pressure is not proper. Maybe the authors wanted to express "exhibit signatures of selection"?

6. L288: is it possible that this difference is due to gene gains in the other lineages?

7. Just suggestions: As clearly mentioned in L272-275, it would be exciting to have morphological evolution from a phylogenetic perspective as future directions. I can foresee that the authors can do a great job on that. Also interesting were the losses of genes whichwere associated with a higher evolution rate for C. chinensis and its relatives, compared with the subgenus Mongogenella. Further studies can be focused on the reasons causing the pattern if this has not been studied.

Grammar or writing mistakes:

L69: existing or extant to replace existent

L87: within the Convolvulaceae

L170: tense

L203: [46]

L222: in

L228: raise => arose or originated

Author Response

Open Review

Comments and Suggestions for Authors

Park et al., sequenced the chloroplast genomes of two Cuscuta species and did nice comparative genomics as well as morphological analysis. The study was nice, and the manuscript was well written.

Response: Thank you for your positive comments. We revised the manuscript according to your suggestions.

Major points:

 L337-340: confusing. First, there was only one phylogeny. Why was GTR+I+G used for two species? Second, it seemed that for such analyses most people may use partitioned phylogenetic reconstruction where different substitution models were given to different partitions of the entire alignment. The authors mentioned that JModelTest was used. Did it mean that the best models of the DNA alignment of each of the 50 genes were determined by JModelTest? If so please show more details in supplemental data. Third, was the third nucleotide in a codon included in analysis? Please correct me if I misunderstood anything.

Response: Thank you for your comments. We had mistaken in sentence. The manuscript was revised (Line 334). In our phylogenic analysis, we used other chloroplast genomes reported. We also used two methods: 1) Maximum likelihood and 2) Bayesian Inference analysis. In this study, ML and BI methods were applied by GTR+I+G from jModeltest results after alignment and manually editing from 50 conserved protein-coding sequences. Additional jModeltest results were shown in below (Table S12). We used 50 conserved protein-coding genes from 19 chloroplast genomes in an attempt to obtain stable results according to the APGIV system. In the case of duplicated genes, we used one gene to protect against biased results. Our analysis was optimized for phylogenic analysis of chloroplast genomes. In particular, we re-analyzed the reconstructed phylogenic tree using MrBayes, with 1,000,000 generations and 1000 replications for ML bootstrap support.

We analyzed the phylogenetic relationships among Cuscuta species with previously reported chloroplast genomes from other species (Ipomoea) to clarify the positions of the Cuscuta species and to detect clustering of species within this family. We identified the phylogenetic relationships between these Cuscuta species and compared them. Moreover, we considered in phylogenic trees with consistence to APGIV system of classification. As a results, chloroplast genome of Cuscuta were dramatically divided in subgenera. We suggest firstly report for phylogenetic trees and morphological, micromorphological traits is consistent. Thus, phylogenetic analysis based on chloroplast protein-coding sequences is useful approach for species identification and classification.

JModelTest result in this study

Table S12 Best-fitting substitution models selection using jModelTest

Model

f(a)

f(c)

f(g)

f(t)

kappa

titv

Ra

Rb

Rc

Rd

Re

Rf

pInv

gamma

AIC

GTR+I+G

0.3

0.18

0.21

0.31

0

0

1.695

2.013

0.254

0.849

2.137

1

0.27

0.87

AIC model selection

Model

-lnL

K

AIC    

delta     

 weight    

cumWeight

GTR+I+G

128183.1683

46

256458.3365

0

0.997641

0.997641

GTR+G

128190.2153

45

256470.4307

12.09412

0.002359

1

GTR+I

128265.7386

45

256621.4773

163.14072

3.74E-36

1

HKY+I+G

129136.7981

42

258357.5963

1899.25974

0.00E+00

1

HKY+G

129143.9543

41

258369.9087

1911.57214

0.00E+00

1

HKY+I

129207.9256

41

258497.8512

2039.5147

0.00E+00

1

SYM+I+G

129477.1116

43

259040.2231

2581.88656

0.00E+00

1

SYM+G

129484.9678

42

259053.9356

2595.59908

0.00E+00

1

SYM+I

129550.8414

42

259185.6827

2727.3462

0.00E+00

1

GTR

129681.2137

44

259450.4273

2992.09078

0.00E+00

1

K80+I+G

130110.5561

39

260299.1122

3840.77564

0.00E+00

1

K80+G

130118.5079

38

260313.0158

3854.6793

0.00E+00

1

K80+I

130182.1891

38

260440.3782

3982.0417

0.00E+00

1

F81+I+G

130210.5895

41

260503.179

4044.84248

0.00E+00

1

F81+G

130217.2105

40

260514.4209

4056.0844

0.00E+00

1

F81+I

130277.8176

40

260635.6352

4177.29868

0.00E+00

1

HKY

130652.8223

40

261385.6446

4927.30808

0.00E+00

1

SYM

131019.6459

41

262121.2917

5662.9552

0.00E+00

1

JC+I+G

131090.6796

38

262257.3592

5799.02268

0.00E+00

1

JC+G

131097.8017

37

262269.6033

5811.2668

0.00E+00

1

JC+I

131159.0792

37

262392.1584

5933.82184

0.00E+00

1

K80

131678.6046

37

263431.2092

6972.87262

0.00E+00

1

F81

131698.7243

39

263475.4485

7017.11198

0.00E+00

1

JC

132612.4124

36

265296.8248

8838.48828

0.00E+00

1

-lnL: negative log likelihod

 K: number of estimated parameters

 AIC: Akaike Information Criterion

 delta: AIC difference

 weight: AIC weight

 cumWeight: cumulative AIC weight

L170-173: Why was the chloroplast genome of Cuscuta different from other angiosperms? Was that what was shown in L221? Could consider removing L170-173, as it was not until L221 that this point was discussed. Also, more reasons may be needed to support this argument. Likewise, in L234, where was the evidence? Importantly, the case for other plants should be detailed.

Response: We revised as your comment. Previous Cuscuta chloroplast genome reports were not considered subgenera level. In this study, we found out chloroplast genome structure, gene contents including sequences variability of Cuscuta species were differ to compare with subgenera (Subg. Monogynella and Subg. Grammica).

L 170 - 173 -> removed

L 221 -> L216 in revised manuscript

L 234 -> L229 in revised manuscript

The paper 10.1093/jxb/ers391 needs to be cited somewhere. The argument raised in L245-247 may not be the first so references are needed.

Response: We cited reference.

Minor points:

1. Figure 2's resolution needs to be improved.

Response: We revised Figures.

2. In Figure 4. BI of 1.0 was shown as "-". This was a bit confusing to me because i) sometimes this symbol indicates not highly supported nodes ii) that "+" and "-" were shown together makes people wonder whether "-" represents negative results. I understand that the authors plausibly wanted to use it to distinguish between bootstrap values and BI. However, I encourage the authors to revise it or alternatively please tell me if there are other good reasons for it or studies presenting it in the same manner.

Response: Figure 4 legend were revised as your comment.

3. Figure S7: Genes whose ka == 0 can be shown as they do not affect the calculation of ka/ks (very strong negative selection). But the authors could keep it as it was for this time.

Response: We revised in Figure S7 as your suggestion.

4. The accession number in Genbank for C. pentagona was different between Table 1 and Table S10.

Response: we revised accession number in Table 1. Table S10 is correct.

5. L59: exhibit selection pressure is not proper. Maybe the authors wanted to express "exhibit signatures of selection"?

Response: We revised as your suggestion.

6. L288: is it possible that this difference is due to gene gains in the other lineages?

Response: Chloroplast genomes of most plants are highly conserved in genome structure, gene content, gene order. Parasitic plant chloroplast genomes can have lower selective pressure due to less reliance on photosynthesis. Gene loss or genome rearrangement were affected in a smaller chloroplast genome size. Orobanchaceae, Broomrape family, Epifagus virginiana, Lathraea squamaria, Aureolaria virginica and Buchnera americana etc. Line 185 in revised manuscript

7. Just suggestions: As clearly mentioned in L272-275, it would be exciting to have morphological evolution from a phylogenetic perspective as future directions. I can foresee that the authors can do a great job on that. Also interesting were the losses of genes which were associated with a higher evolution rate for C. chinensis and its relatives, compared with the subgenus Mongogenella. Further studies can be focused on the reasons causing the pattern if this has not been studied.

Response: Thank you very much for your kind words about our work and future direction. According to the phylogenetic classification of the genus Cuscuta, number of style and IFS type are informative diagnostic characters for distinguishing subgenera Grammica and Monogynella. Moreover, testa ornamentation is also consistent with the subgeneric classification. Although these reproductive morphological characteristics are consistent with molecular phylogeny, it is difficult to say the possibility of the correlation between morphological traits (phenotypes) and molecular sequences (genotypes) based on our chloroplast genomic results only. Following your valuable comments, we will try to investigate association the phenotypic variation and gene loss using Evo-Devo approach and transcriptome analysis etc.

Grammar or writing mistakes:

L69: existing or extant to replace existent

L87: within the Convolvulaceae

L170: tense

L203: [46]

L222: in

L228: raise => arose or originated

Response: We revised as your comment.

Reviewer 2 Report

This is very well done work, which is also very well written. The objectives are clear, as the methodology used is correct, and the conclusions useful. I recommend it to be published. 

However, there are some problems with Figures. In Figures 2, 3 and S5, the features and names and numbers are not visible at all, and in this form, the figures are useless. Also, being the main point of the paper to show a comparison of Cuscuta japonica and C. chinensis, I would ask to edit the figures so as the both taxa are shown next to each other. In Figure S3, it is not clear which part belongs to which species. Something seems to be missing here.

Some minor comments:

Line 13 please use “High-throughput Sequencing‎” -there are at least 2 “new generation sequencing techniques“ after the old Sanger sequencing.

Line 245-246  “This suggests that IRs in Cuscuta plastid genomes underwent more contraction than those in chloroplast genomes of other angiosperms, indicating an independent evolutionary process in response to the environment” - how do you deduce this? You have no specific experiment or modeling method to verify this, neither do you provide a citation. Please clarify.

Author Response

Open Review

Comments and Suggestions for Authors

This is very well done work, which is also very well written. The objectives are clear, as the methodology used is correct, and the conclusions useful. I recommend it to be published. 

However, there are some problems with Figures. In Figures 2, 3 and S5, the features and names and numbers are not visible at all, and in this form, the figures are useless. Also, being the main point of the paper to show a comparison of Cuscuta japonica and C. chinensis, I would ask to edit the figures so as the both taxa are shown next to each other. In Figure S3, it is not clear which part belongs to which species. Something seems to be missing here.

Response: Thank you for your comments. Figure 2 was shown completed chloroplast genomes of C. japonica and C. chinensis, respectively. We also represented Figure 3 and Figure S5 to compare with reported other Cuscuta chloroplast genome for genome structure variation among seven Cuscuta species. We changed Figures to see better. Figure legend, name, features are confirmed. We did not found some problems for Figures. Compressed zip file was attached in revision version.

Some minor comments:

Line 13 please use “High-throughput Sequencing‎” -there are at least 2 “new generation sequencing techniques“ after the old Sanger sequencing.

Response: We revised in the Introduction. Line 47 - 48

Line 245-246  “This suggests that IRs in Cuscuta plastid genomes underwent more contraction than those in chloroplast genomes of other angiosperms, indicating an independent evolutionary process in response to the environment” - how do you deduce this? You have no specific experiment or modeling method to verify this, neither do you provide a citation. Please clarify.

Response: We cited reference.

Braukmann, T.; Kuzmina, M.; Stefanovic, S. Plastid genome evolution across the genus Cuscuta (Convolvulaceae): two clades within subgenus Grammica exhibit extensive gene loss. J. Exp. Bot. 2013, 64, 977-989, doi:10.1093/jxb/ers391.